# Biochemical Reactions and Their Biological Contributions in Honey

**DOI:** 10.3390/molecules27154719

**Published:** 2022-07-23

**Authors:** Wed Mohammed Ali Alaerjani, Sraa Abu-Melha, Rahaf Mohammed Hussein Alshareef, Badriah Saad Al-Farhan, Hamed A. Ghramh, Badria Mohammed Abdallah Al-Shehri, Majed A. Bajaber, Khalid Ali Khan, Munira M. Alrooqi, Gad Allah Modawe, Mohammed Elimam Ahamed Mohammed

**Affiliations:** 1Department of Chemistry, Faculty of Science, King Khalid University, Abha 61413, Saudi Arabia; 441813276@kku.edu.sa (R.M.H.A.); mb@kku.edu.sa (M.A.B.); 2Department of Chemistry, Faculty of Girls for Science, King Khalid University, Abha 61413, Saudi Arabia; sabomlha@kku.edu.sa (S.A.-M.); bfrhan@kku.edu.sa (B.S.A.-F.); balshehre@kku.edu.sa (B.M.A.A.-S.); 3Unit of Honeybee Research and Honey Production, King Khalid University, Abha 61413, Saudi Arabia; hamedes@hotmail.com (H.A.G.); khalidtalpur@hotmail.com (K.A.K.); 4Research Center for Material Science, King Khalid University, Abha 61413, Saudi Arabia; 5Department of Biology, Faculty of Science, King Khalid University, Abha 61413, Saudi Arabia; 6Department of Chemistry, Faculty of Applied Science, Umm Al Qura University, Makkah 24382, Saudi Arabia; mmrooqi@uqu.edu.sa; 7Biochemistry Department, Faculty of Medicine and Health Sciences, Omdurman Islamic University, Omdurman 14415, Sudan; gadobio77@hotmail.com

**Keywords:** glucose oxidase, catalase, carbohydrates, amino acids, dicarbonyl compounds, HMF

## Abstract

Honey is known for its content of biomolecules, such as enzymes. The enzymes of honey originate from bees, plant nectars, secretions or excretions of plant-sucking insects, or from microorganisms such as yeasts. Honey can be characterized by enzyme-catalyzed and non-enzymatic reactions. Notable examples of enzyme-catalyzed reactions are the production of hydrogen peroxide through glucose oxidase activity and the conversion of hydrogen peroxide to water and oxygen by catalase enzymes. Production of hydroxymethylfurfural (HMF) from glucose or fructose is an example of non-enzymatic reactions in honey.

## 1. Introduction

Honey is a sweet honeybee product which is mainly composed of sugars and water. Biomedical activities of honey are mostly due to its minor components, which include proteins, amino acids, organic acids, dicarbonyl molecules, hydrogen peroxide, phenolic acids, flavonoids, and enzymes [1,2,3]. Honey enzymes originate from three major sources: plant nectars and secretions, honeybees, and excretions of plant-sucking insects. Biochemical reactions can be divided to two types: enzyme-catalyzed and non-enzymatic reactions [4]. Enzyme-catalyzed reactions in honey are known to affect its quality and biological activities [5,6,7]. Enzymes present in honey include the diastase, invertase, glucose oxidase, catalase, glucosylceramidase, α-amylase, α-glucosidase, β-glucosidase, and proteases [6,8]. 

Enzymatic reactions in honey include the conversion of oligosaccharides and disaccharides (sucrose and maltose) to glucose and fructose by diastase and invertase enzyme activity. Glucose is converted to gluconic acid and hydrogen peroxide by glucose oxidase. Moreover, hydrogen peroxide is degraded to water and oxygen by catalase enzymes. Honey samples which exhibit high catalase activity are low in hydrogen peroxide [9]. Honey samples with high hydrogen peroxide concentrations are known to be useful for the treatment of wounds, and are characterized by a high activity of glucose oxidase and low activity of catalase [10]. Furthermore, honey and other honeybee products are characterized by enzyme-catalyzed reactions such as the proteases, glucosylceramidase and acid phosphatase [6,11]. 

Classic non-enzymatic reactions in honey are the production of HMF from glucose or fructose, the formation of diacarbonyl molecules such as methylglyoxal from the dihydroxyacetone of nectar, transglycosylation, and the formation of amino sugars (Maillard reaction) [12,13,14,15].

In this review, we tried to confine the enzymatic and non-enzymatic reactions or pathways that occur in honey. The biochemical reactions or pathways in honey play a key role in determining the quality and biological activity of honey samples.

This review article illustrates the biochemical reactions and their substrates, intermediates and products that occur in honey. The intermediates and products of the biochemical reactions are responsible for the biological activities of honey, such as the hydrogen peroxide, methylglyoxal, organic acids, HMF and melanoidins. Moreover, to the best of the authors’ knowledge, this article is the first review concerning the enzymatic and non-enzymatic biochemical reactions in honey.

## 2. Enzymatic Reactions in Honey

### 2.1. Production and Degradation of Hydrogen Peroxide

Hydrogen peroxide is produced as a product of glucose oxidation by glucose oxidase and by the non-enzymatic autoxidation of polyphenols [16,17]. Glucose oxidase is secreted by honeybees; some studies have reported its production by plants, honey fungi, yeast, and bacteria [18,19,20,21]. Glucose oxidase catalyzes the conversion of glucose to gluconic acid and hydrogen peroxide, using molecular oxygen and vitamin B2 as cofactors (FAD) (Figure 1).

Honey hydrogen peroxide is degraded by enzymatic and non-enzymatic reactions. The major catabolic enzyme of hydrogen peroxide is Fe^+3^-dependent catalase, which catalyzes the conversion of two H_2_O_2_ molecules to two water molecules and one oxygen molecule. Plant nectar and the microorganisms in honey are the major sources of catalase in honey [16,22,23]. Other metal-containing enzymes (metalloenzymes) are capable of converting hydrogen peroxide to water following a different mechanism of action. Metalloenzymes include peroxidases and superoxide dismutase (Figure 1) [24].

Non-enzymatic reactions responsible for the degradation of hydrogen peroxide are the vitamin C and Fenton reactions. Vitamin C (hydro ascorbic acid) donates two protons and two electrons to hydrogen peroxide, reducing it to water and oxygen. It is reported that the addition of vitamin C to honey samples leads to decreased hydrogen peroxide levels, which reinforces the suggestion that vitamin C degrades hydrogen peroxide [16,25]. The second non-enzymatic reaction that depletes hydrogen peroxide from honey is the Fenton reaction. Fenton reactions involve the reaction of hydrogen peroxide with Fe^+2^ and/or polyphenols to produce hydroxide ions, oxygen, and hydroxyl and superoxide radicals. The hydroxyl and superoxide free radicals contribute to the antibacterial activity of honey because of their powerful damaging effect on the bacterial cells and DNA (Figure 1) [26,27].

Hydrogen peroxide generation in honey is influenced by its colloidal structure, which is due to the interactions between its macromolecules. Macromolecules in honey involved in the formation of the colloidal structure include oligo-sugars, proteins, and polyphenols. Colloidal particles of honey are composed of complexes of proteins, polyphenols, and melanoidins. The colloids of honey are compact and stable, with multiple layers. Colloidal honey samples are characterized by the generation of hydrogen peroxide, and antibiotic and antioxidant activities. Moreover, the colloidal structure of honey increases in dark and medium-color honeys, whereas it is not a character of the light color honeys [16,28].

Most of the antibacterial activity of honeys is due to the hydrogen peroxide concentration, high sugar concentration, and the low pH, due to the content of organic acids in honey, such as gluconic acid. Hydrogen-peroxide-producing honeys are used in wound and burn dressings, either alone or in combination with other medicines such as calcium alginate [16,28,29] (Table 1).

### 2.2. Proteases

Proteases are responsible for the degradation of proteins to produce amino acids and short peptides according to the type of protease. According to their substrate preference, proteases are classified as endopeptidases and exopeptidases [69,70]. Unifloral honey samples are reported to contain serine protease enzymes, which contribute to their quality and biological activities [6]. A Croatian study reported the presence of three serine proteases in honey: trypsin, chymotrypsin, and elastase [37]. Trypsin cleaves the peptide bonds formed by the carboxyl groups of arginine and lysine [71]. The sites of chymotrypsin cleavage are peptide bonds formed by the carboxyl groups of aromatic amino acids (tyrosine, phenylalanine, and tryptophan) and leucine [72]. Elastase is associated with breakdown of the peptide bonds formed by the short aliphatic amino acids such as glycine, alanine, and valine [73]. The major products of honey proteases are short peptides that function as antioxidants, antitumor, and antimicrobials, and are used as weight loss inducers (Figure 2). Alaerjani et al. (2021) published an article which proved the presence of short peptides in honey samples from Saudi Arabia. They analyzed five honey samples using LC–MS and concluded that short peptides in honey samples are of floral origin and storage-dependent [30] (Table 1).

Amino acids and their sequences in short peptides are responsible for their activity. Short peptides that contain cysteine, methionine, tyrosine, lysine, histidine, and tryptophan are known to act as antioxidants [31,32,33]. Short peptides rich in hydrophobic amino acids, such as glysine, alanine, valine, leucine, and isoleucine, are active antimicrobial peptides because of their ability to disrupt the plasma membrane of the microbes [30,34]. Short and cyclic peptides are active as antitumor, antimicrobial, antioxidant, and weight loss inducers, according to their amino acid contents and sequences [35]. Assessments of the amino acid contents, sequences, and concentrations of short peptides in food are carried out using LC–MS techniques [30,74] (Table 1).

## 3. Enzymes in Honey as Quality Parameters and Indicators for Floral Origin, Honeybee Disease and the Presence of Yeast

### 3.1. Diastase and Invertase

Diastase (α- and β-amylases) and invertase (α-glucosidase) are hydrolytic enzymes secreted by honeybees to help ripen nectar to produce the mature honey. Diastase is responsible for the conversion of nectar polysaccharides (amylose) to glucose, whereas invertase converts the sucrose of the nectar into fructose and glucose. Diastase enzymes are included as honey quality parameters and indicators for assessing honey storage conditions in all honey standards [1]. Invertase is not adopted as a honey quality parameter; however, it is suggested to be a useful quality parameter in the European standards for honey [36]. Although diastase and invertase originate from honeybees, they can be used as indicators for honey floral origins because the concentration of the substrates affects the enzyme activity [75]. Some studies used the activities of diastase and invertase as quality parameters to predict the floral origin of honey samples through adopting chemometric analysis (principal component analysis (PCA) and clustering analysis) [76,77,78] (Table 1).

### 3.2. β-Glucosidase

β-glucosidase is an enzyme responsible for the breakdown of β-glucosidic bonds, such as those of cellulose and cellobiose. The β-glucosidase enzyme can be isolated from the ventriculus, honey sac, and hypopharyngeal glands of the Apis malifera [79]. Honey samples have been reported to exert β-glucosidase activity [77]. 

### 3.3. Glucosylceramidase (Glucocerebrosidase or Ceramide Glucosidase)

Glucosylceramidase catalyzes the hydrolysis of glucosylceramide (glycolipid) to glucose and ceramide [80]. Glucosylceramidase is reported to be present in buckwheat honey. Deficiencies of glucosylceramidase in insects lead to the accumulation of glucosylceramide, which is associated with abnormalities in memory and movement. Moreover, deficiencies of glucosylceramide in insects and honeybees lead to neurodegeneration and a shorter lifespan (Figure 3) [8]. 

### 3.4. Acid Phosphatase

Acid phosphatase is a hydrolytic enzyme which catalyzes the production of inorganic phosphate from organic phosphate. Organic phosphates which act as substrates for acid phosphatase include ATP, phosphatidic acid, and phosphate monoester [81,82,83]. 

The sources of acid phosphatase in honey are pollens, nectar, and yeast. The enzyme could be used as a marker for the botanical origin of honey and as an indicator of honey fermentation [37,38,39] (Table 1). The activity of acid phosphatase in honey is affected by the pH and climate conditions [39].

## 4. Non-Enzymatic Reactions in Honey

### 4.1. Non-Enzymatic Trans-Glycosylation

The formation of di- and oligosaccharides in honey is proved to be through non-enzymatic trans-glycosylation. Trans-glycosylation reactions in honey are promoted by the high concentration of sugars, low moisture percentage, and acidic environment. The disaccharides which are formed through trans-glycosylation are maltose, isomaltose, inulobiose, sophorose, and gentiobiose, whereas the oligosaccharides include 1-kestose, melezitose, and panose [13,84]. Oligosaccharides are classified to four classes according to their structural units. The four oligosaccharides are maltoligosaccharides, fructooligosaccharides, chitanooligosaccharides, and galactooligosaccharides. Fructooligosacchrides constitute 0.75% of honey [85]. Fructooligosaccharides are used as artificial sweeteners, and are characterized by their low caloric values [40]. Fructooligosaccharides with calcium supplements are useful in increasing the bone mineral density in postmenopausal women [41] (Table 1).

The oligosaccharides of honey are active as prebiotic molecules. Prebiotics are non-digestible foods which have functional effects on the gastrointestinal tracts of animals through stimulating the growth of intestinal flora (bifidobacteria and lactobacilli). The intestinal flora is capable of fermenting the oligosaccharides, leading to a decrease in pH, the production of short-chain fatty acids, and reductions in fat absorption and ammonia production [42,43] (Table 1).

### 4.2. Production and Degradation of Dicarbonyl Compounds

Dicarbonyl compounds are formed in thermally processed foods from the oxidation and degradation of sugars, the Maillard reaction, degradation of lipids, and degradation of vitamin C [86]. Dicarbonyl compounds found in honey samples of different floral origins include glyoxal, methylglyoxal, glucosone, 3-deoxyglucosone, 2,3-butanedione, 3-deoxypentulose, 1,4-dideoxyhexulose, and 3,4-dideoxyglucoson-3-ene (3,4-DGE) [87,88]. Adams et al. (2009) proved that the methylglyoxal of Manuka honey originated from dihydroxyacetone from the flowers of the Manuka tree (Leptospermum scoparium). Dihydroxacetone is non-enzymatically converted to methylglyoxal at a temperature of 37 °C [89]. Dihydroxyacetone of the Manuka tree nectar is mostly obtained from dihydroxyacetone phosphate (the glycolysis intermediate) through hydrolysis reactions. The conversion of dihydroxyactone phosphate to dihydroxyacetone may occur enzymatically through dihydroxyacetone phosphatase, such as reactions in the Corynebacterium glutamicum [90], or through non-enzymatic hydrolysis (Figure 4).

Dicarbonyl molecules are catabolized through variable non-enzymatic reactions to volatile aroma compounds and food-browning molecules [44]. Small dicarbonyl molecules (glyoxal and methylglyoxal) can react with amino groups of free amino acids such as methionine and leucine to form methional and methylbutanal, respectively. Methional and methylbutanal contribute to food flavors and odorants [45]. Moreover, the dicarbonyl molecules can bind the amino and guanido groups of lysine, nucleosides, and arginine residue on protein or nucleosides to form complexes such as advanced glycation end products (AGEs) and nucleoside AGEs [46,47]. The presence of sodium chloride in foods leads to the conversion of dicarbonyl molecules to furfurals, whereas reactions of the dicarbonyl compounds with phenolic molecules form complexes responsible for food browning, such as melanoidins [48,49]. 

Dicarbonyl molecules are major contributors to the antibacterial activity of honey, such as methylglyoxal of Manuka honey [50,51]. High blood levels of dicarbonyl compounds are reported to be associated with diabetes, uremia, and negative effects on the blood vessels, whereas glyoxal exhibits tumorigenic activity [44,52]. Some studies have reported that methylglyoxal exhibits tumorigenic and antitumor activities. Antitumor activities of methylglyoxal include the induction of cancer cell apoptosis and inhibition of the proliferation and invasion of colon cancer, leukemia, and breast cancer cells in vitro and in vivo [44]. The tumorigenic activity of methylglyoxal is possibly through different modes of actions, such as the depletion of ATP through activating glycation reactions on specific arginine residues on ATP-producing enzymes, the glycation of nucleosides on DNA, and the glycation of histones [44]. The presence of methylglyoxal in foods with creatine or ascorbic acid supports its anticancer activity [44] (Table 1).

### 4.3. Production and Degradation of HMF

HMF is a six-carbon cyclic compound with two functional groups: aldehyde and hydroxymethyl. HMF is produced from the non-enzymatic degradation of glucose or fructose. HMF is considered to be an intermediate product of the Maillard reaction (browning of food). Glucose is converted to fructofuranose through isomerization reactions, and fructofuranose undergoes three dehydration reactions and enolization to produce HMF (Figure 5) [14,91,92].

The formation of HMF in honey is induced by the pH, acidity, minerals, and hexose sugars. Storage and processing conditions are known to increase the production rate of HMF, i.e., a high temperature and long storage duration (even at room temperature) [93,94,95,96]. The storage of honey at low temperatures (below 25 °C) does not facilitate the excessive production of HMF, which exceeds the range of international honey standards [97,98].

HMF is degraded non-enzymatically to produce formic and levulinic acids, in addition to soluble polymers and insoluble humins [92,99] (Figure 5).

Two studies have suggested that high HMF concentrations are not an indication for honey storage at a temperature range of 21–50 °C [100,101]. Fallico et al. (2008) proved that the degradation rate of HMF in citrus and chestnut honey is greater than its production rate at temperatures between 25 °C and 50 °C [100]. Moreover, Khan et al. (2021) reported that freshly harvested honey samples are characterized by high HMF concentrations compared with honey stored at room temperature (21–36 °C) for one year [53]. The findings of Fallico et al. (2008) and Khan et al. (2021) illustrated that stored honey, or honey samples treated at temperatures between 21 °C and 50 °C, are characterized by low HMF concentrations [100,101]. Consequently, Fallico et al. (2008) suggested the removal of HMF from the honey standards because it is not an indicator for honey storage or treatment [100]. The findings of Fallico et al. (2008) and Khan et al. (2021) contradict the well-established fact that HMF production in honey increases during storage and heat processing. The major scientific fact behind their findings is the production and degradation rates of HMF in a temperature range of 21–50 °C.

HMF in honey threatens the life of honeybees [53,54]; moreover, it has negative and positive effects on human health. Negative effects of HMF on human health include hepato-renal toxicity, decreased glutathione levels in the cells, inductions of neoplastic changes, and irritation of the mucous membranes of the upper respiratory tract, eyes, and skin [55,56,57]. Positive effects of HMF on human health include antioxidant, anti-carcinogenic, anti-allergenic, and anti-hyperuricemic activities [58,59,60] (Table 1).

The toxic level of HMF for humans is not yet clarified, because the metabolism of HMF is health-condition-dependent. However, humans can ingest from 30 to 150 mg of HMF daily from different food sources, and it is reported that HMF is completely cleared from human bodies within 24–48 h, even if taken at a level of 240 mg/day [91]. From the researched literature, it is obvious that honey is very safe for human consumption.

### 4.4. Maillard Reaction in Honey

The Maillard reaction is associated with reactions of sugars and amino acids in water. It is responsible for food browning during heating processes. The Maillard reaction is defined as an interaction between the carbonyl groups of sugars and groups of amino acids, proteins, or other nitrogenous compounds, leading to the production of brown compounds known as melanoidins [102]. The Maillard reaction has three stages: early, advanced, and final stage. The early stage of the Millard reaction involves a combination of sugars and amino groups, which leads to the formation of Amadori products (complexes of sugars and amino acids) such as N-(1-Deoxy-1-fructosyl) phenylalanine, which has recently been found in heated acacia honey samples [103]. In the advanced or intermediate stage, the Amadori products are broken down to different compounds including the amino aldoses and ketoses. The amino aldoses are further degraded to form deoxy dicarbonyls and enediols, while amino ketoses are converted to 2-amino-2-deoxy-aldoses through Heyns rearrangement. The intermediate-stage reactions are acid-catalyzed [104]. The final stage of the Millard reaction is associated with the formation of melanoidin polymers, which are formed from sugar fragments, amino products, polyphenols, quinones, and proteins. Generally, melanoidins are associated with food browning [14,105,106] (Figure 6).

Brudzynski et al. (2013) stated that the storage of honey at different temperatures decreased the concentration of honey proteins and led to the formation of complexes of proteins–polyphenols and proteins–quniones [106]. 

Hellwig et al. (2017) found that samples of Manuka honey were characterized by high concentrations of Maillard reaction products, such as methylglyoxal-derived hydroimidazolone 1 (MG-H1) and carboxyethyllysine (CEL) [15].

The Maillard reaction is induced by different factors such as the water activity, initial pH, amino group, oxygen, sugar temperature, and heating time [61]. Many studies have reported that Maillard reaction products act as antioxidant and antibacterial agents [14,62,63,64] (Table 1).

### 4.5. Caramelization in Honey

Caramelization is the heating of any sugar using high temperatures (≥180 °C) until its color is changed to brown–black. It is a non-enzymatic process involving the dehydration of sugars under dry or concentrated solutions. Caramelization produces solid, semi-solid, and liquid products, with colors ranging from light brown to medium and dark. The products of caramelization depend on the temperature used. To obtain a light brown color, the sugar or food is heated to 180 °C; a medium color can be achieved by heating at 181–187 °C; and a dark color is obtained through heating at 188–204 °C [107,108,109].

Rahardjo et al. (2020) obtained light, medium, and dark caramel colors of honey by heating at 107 °C, 180–182 °C, and 188–190 °C, respectively [107]. The products of the caramelization reaction included deoxyosones, furan and pyran derivatives, HMF, hydroxydimethylfuranone (HDF), and hydroxyacetilfuran (HAF). The products of caramelization reactions contribute to the color, aroma, and flavor of honey, in addition to exerting antioxidant activity [65,66,67,68] (Table 1).

## 5. Conclusions

This review summarized, for the first time, the biochemical reactions in honey, including enzymatic and non-enzymatic reactions. The substrates, intermediates, and products of the biochemical reactions in honey are major contributors to the biological activities of honey.

## Figures and Tables

**Figure 1 molecules-27-04719-f001:**
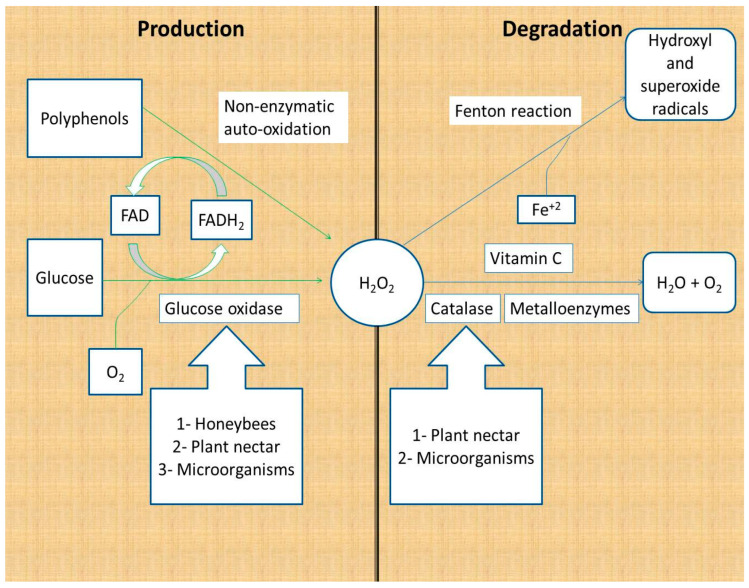
Production and degradation of hydrogen peroxide in honey. Hydrogen peroxide is produced from glucose by the action of glucose oxidase and non-enzymatically by polyphenols. Hydrogen peroxide is degraded to water and oxygen by enzymes and vitamin C, whereas it is degraded to hydroxyl and superoxide radicals through the Fenton reaction [16,17,18,19,20,21,22,23].

**Figure 2 molecules-27-04719-f002:**
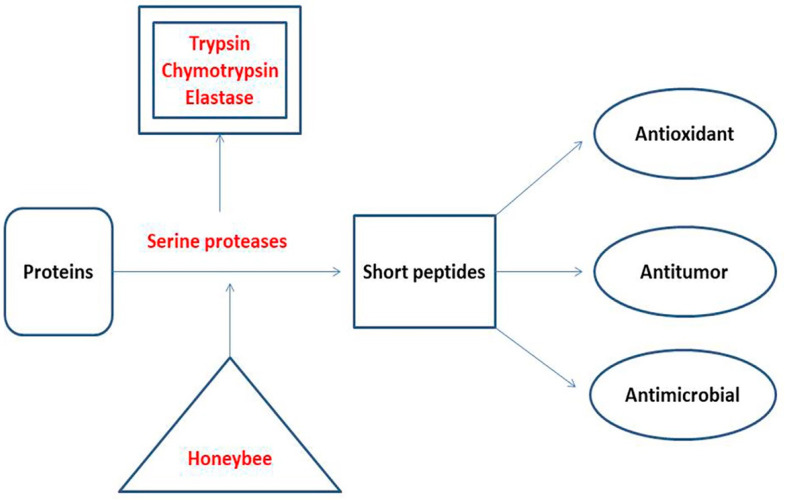
The proteases in honey: their types, origin, substrate, products, and their contribution to the biological activities of honey [6,30,37,69,70,71,72,73].

**Figure 3 molecules-27-04719-f003:**
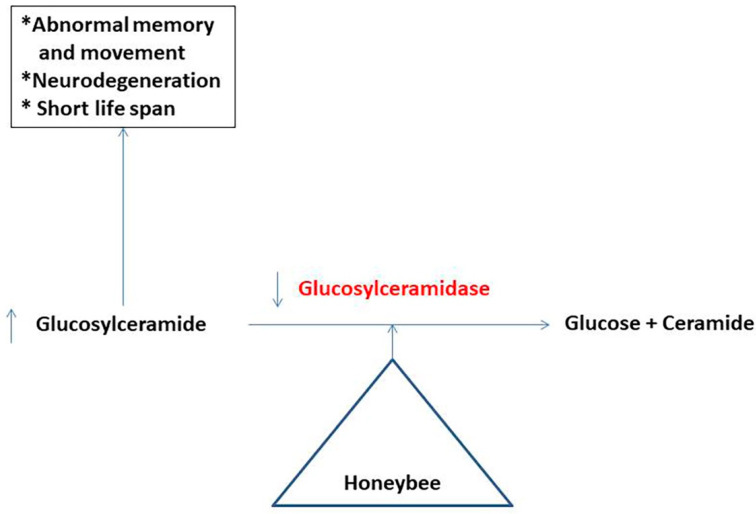
Glucosylceramidase in honey and its effect on the honeybee’s health [8,80].

**Figure 4 molecules-27-04719-f004:**
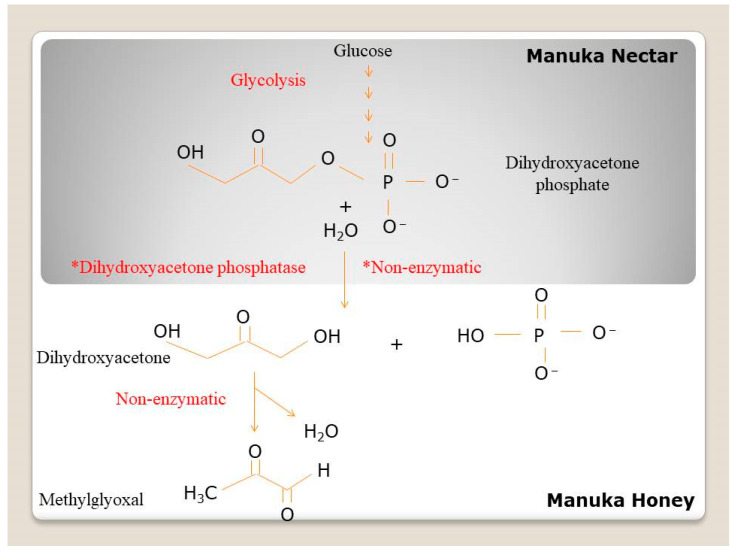
Production of methylglyoxal in Manuka honey [89].

**Figure 5 molecules-27-04719-f005:**
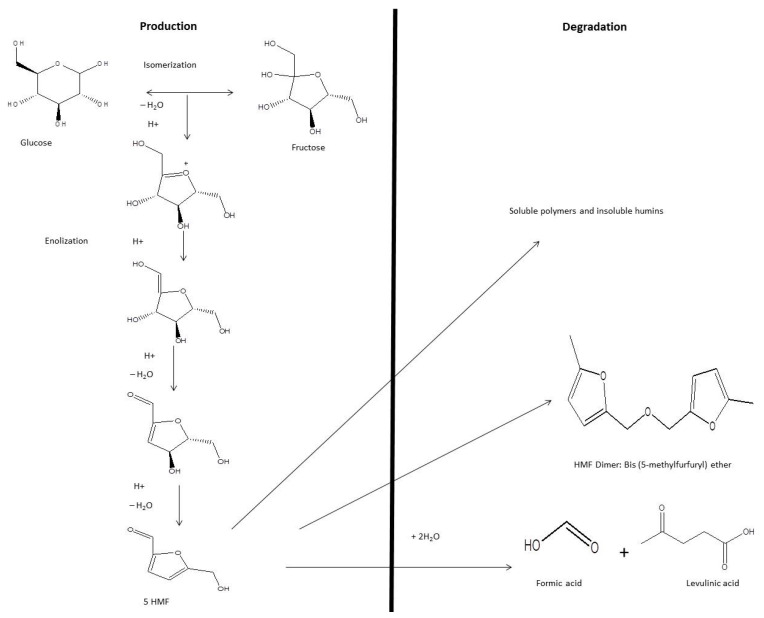
Production and degradation of HMF in honey [14,91,92].

**Figure 6 molecules-27-04719-f006:**
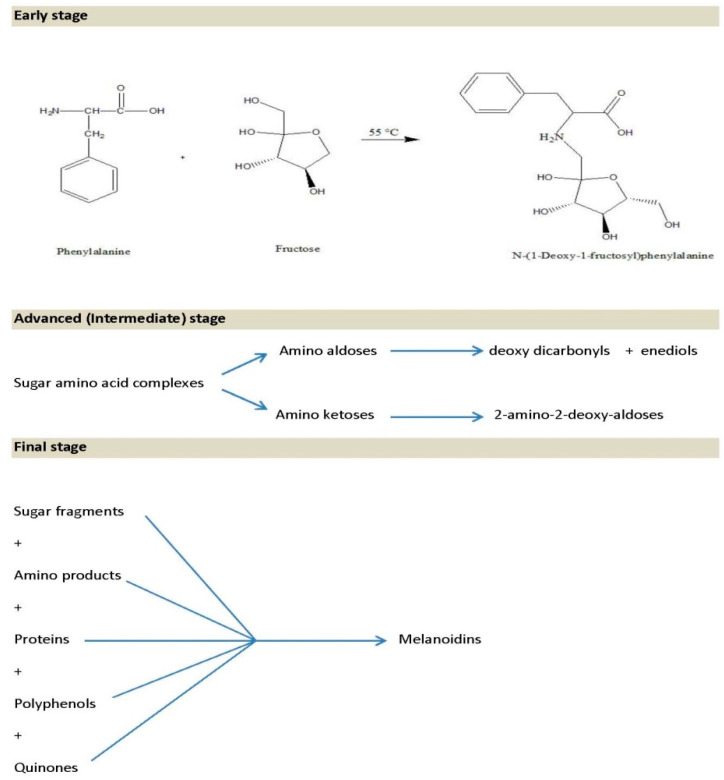
Stages of the Maillard reaction in honey [14,103,104,105,106].

**Table 1 molecules-27-04719-t001:** Honey reactions and their biological activities.

	Reaction	Enzyme	Products	Biological Activity	Ref.
1	Production of hydrogen peroxide	Glucose oxidase	Hydrogen peroxide and gluconic acid	Antibacterial and wound and burn dressings	[16,28,29]
2	Production of short peptides	Proteases	Short peptides	Antimicrobial, antioxidant, antitumor and weight loss inducers	[30,31,32,33,34,35]
3	Degradation of amylose	Diastase	Glucose and maltose	Honey quality parameter that indicates storage conditions	[1]
4	Degradation of sucrose	Invertase	Glucose and fructose	Indicator for honey storage conditions	[36]
5	Degradation of organic phosphates	Acid phosphatase	Inorganic phosphate	Marker of honey floral origin and indicator of honey fermentation	[37,38,39]
6	Trans-glycosylation	Non-enzymatic	Disaccharides Oligosacharides	Artificial sweeteners, increase bone mineral density in postmenopausal women (fructooligosaccharides), and classified as prebiotic molecules	[40,41,42,43]
7	Production and degradation of dicarbonyls	Enzymatic (dihydroxyacetone phosphatase) and non-enzymatic	Dicarbonyls (glyoxal, methylglyoxal and 3-deoxyglucosone), methional and methylbutanal, AGEs and nucleoside AGEs and melanoidins	1-Antibacterial, antitumor, antioxidant and contribution to the honey color, flavor and odorant. 2-High level of dicarbonyl molecules are reported to be with some toxicity to the humans such as tumerigenic, negative impact on blood vessels and induction of diabtes and uremia	[44,45,46,47,48,49,50,51,52]
8	Production and degradation of HMF	Non-enzymatic	HMF, formic and levulinic acids, soluble HMF polymers and insoluble humins	Threatening honeybee life, human hepatorenal toxicity, induction of neoplastic changes, and irritation of mucous membranes HMF has positive impacts on human health, such as antioxidant, anti-carcinogen, anti-allergenic, and anti-hyperuricemic activities	[53,54,55,56,57,58,59,60]
9	Maillard reaction	Non-enzymatic	Complexes of sugars and amino acids, amino aldoses and ketoses, dicarbonyls, enediols, 2-amino-2-deoxy-ald-oses and melanoidins	Antibacterial and antioxidant	[14,61,62,63,64]
10	Caramelization	Non-enzymatic	Deoxyosones, furan and pyran derivatives, HMF, hydroxydimethylfuranone (HDF) and hydroxyacetilfuran (HAF)	Contribution to the color, aroma, and flavor of honey and antioxidants	[65,66,67,68]

## Data Availability

All relevant data are presented in review article.

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
