# Peer review of "Biochemical Reactions and Their Biological Contributions in Honey"

_molecules, 2022, doi:10.3390/molecules27154719_

Round 1

Reviewer 1 Report

Manuscript ID:  molecules- 1803874

Type:  Review

Title:  Enzyme-catalyzed and non-enzymatic reactions in honey and their contributions to the biological activity of honey

The manuscript with entitle of “Enzyme-catalyzed and non-enzymatic reactions in honey and their contributions to the biological activity of honey” submitted by Wed Mohammed Ali Alaerjani et al. to Molecules as a Review (ID: molecules-1803874).  In this review, the authors attempted to combine and descript the enzymatic and the non-enzymatic reactions or “pathways” that occur in honey as also shown in the title of the manuscript. According to the authors’ statement, biochemical reactions or pathways in honey may play a major role in determining the honey quality and biological activity.

Overall, the topic was interesting but the manuscript was neither in details nor comprehensive. After reading through the review, I think there occur many concerned issues remained to be fully addressed. Recently, the journal “Molecules” got an increased IF of 4.927 thereby the publishing criterion on this particular journal should be strict. I would suggest this paper as major revision, but encouraging the authors to revise the contents and increase the biological information in the manuscript.

Major issues:

1. The title of the manuscript is not concise. It does not fit with the contents. As you can see, the title was about “two types of reactions and their biological contributions in honey”. I thought the review paper would well focus on this point. Well, I am confused by the description of enzymes instead of reactions in the whole main texts. I wonder if the authors may revise the current title a little bit or not, in order to get a proper title.

2. In the abstract, the authors introduced honeybee honey first for containing enzymes inside, and then described that enzyme-catalytic reaction by glucose oxidase to produce hydrogen peroxide and then degrade hydrogen peroxide afterwards by catalase, non-enzymatic reaction each in honey to generate HMF. What is the significance or vital meaning for listing these examples?

3. Line 62, 2.1. Metabolism of hydrogen peroxide  

Maybe you are right. But I do not think this subtitle is suitable here. Obviously, the authors talked about the production and degradation of hydrogen peroxide. Are the generations and the conversions of hydrogen peroxide in honey therefore the metabolism?

4. I agree that short peptides may possess antioxidant, anti-tumor and anti-microbial activities. However, the sequence and the structure of peptides matter a lot. Sequences and quantifications of the short peptides were not presented in the review.

5. The authors mentioned that “Other metal containing enzymes (metalloenzymes) are capable of converting the hydrogen peroxide to water and oxygen following the same mechanism of action of the catalase enzyme.”, Are you sure that both peroxidases and SOD truly obey the same catalytic mechanism of action of the catalase?  

6. Regarding the honey and its biological activities, not too much information can be learned from reading this paper. Is there any table with a list of bioactivities?

7. A simple direct question: Regarding the various enzymes such as glucosylceramidase, glucose oxidase, perdases, superoxide dismutases, catalase, serine proteases, elastase, aiastase (α and β- amylases) and invertase (α- glucosidase) etc., present in honey but from different sources, are they conserved or very similar, or shared the very same catalytic residues or completely identical in primary structure?

8. The current form of this paper is just a little bit superficial for wider readers of “Molecules”. So, what is the impressive knowledge or something brand-new for people to know, or some take home message to digest later on? I wish the authors might highlight the importance and necessity of publishing this review paper to all.

Minor issues:

1. Line 24, in the abstract,  “…enzyme catalyzed reactions…” should be revised to “…enzyme-catalyzed reactions…”.

2. Line 77, “Fe+3” should be revised to “Fe+3” .

3. Line 78, “H2O2” should be revised to “H2O2” . Actually, you knew how to write the correct one as you shown in Figure 1.

4. Lines 80, “…metal containing enzymes” should be revised to“metal-containing enzymes

5. Line 90, “Fe+2” should be revised to “Fe+2” . Actually, you knew how to write the correct one as you shown in Figure 1.

6. The fashions vary from Figure 1 to Figure 4.  What is the use for applying this diversified format? 

Finally, I would recommend this manuscript to be Major Revision.

Author Response

Dear 

Thank you very much for your valuable comments. I have strictly responded to all of your comments

Reviewer 2 Report

This manuscript attempts to describe reactions that take place in honey that contribute to its biological activity. However, several aspects need to be addressed before this manuscript can be published.

At first, the manuscript needs reconstruction. Chapter 3 refers to honey enzymes. This chapter should be right after the Introduction and contain all enzymes reported in honey, their role in honey formation, honeybees, and honey quality and legislation.

Also, since the title includes “their contributions to the biological activity of honey”, this should be an important part of this manuscript. Even though there are scattered references in the manuscript, I would expect a whole chapter, with the appropriate literature, highlighting how the products of these reactions contribute to honey biological activity.

Also, a lot need to be changed in the chapter regarding HMF. At first, Figure 5 should be changed to include fructose as well. Since HMF is formed both from glucose and fructose, this figure should change. Ref {68} has an appropriate scheme, while also you can find one in Capuano and Fogliano, 2011 (doi:10.1016/j.lwt.2010.11.002). In addition, fructose has higher reactivity than glucose (Islam et al., 2013; doi:10.1002/jat.2952). Furthermore, lines 247-251 contain statements that lack sufficient research support. Firstly, it is naïve to suggest that the more HMF a honey contains, the fresher it is!! There are several reports of zero HMF in fresh honeys, like chestnut. Also, there is scientific evidence to support that storage increases HMF in honey. In addition, heat treatment of honey increases HMF when temperature is high (e.g. higher than 55 οC). Honey postharvest processing often uses temperatures as high as 80 οC. I suggest you remove lines 247-250 as this cannot be supported.

Regarding caramelization (Section 4.5.) I wonder why would one caramelize honey? Reference 89 is not about honey, but model systems. Caramelisation requires severe heat treatment and almost never uses honey as a substrate due to its high price. In my opinion, this does not belong to a review paper about reaction in honey.

Finally, section 5 is not a conclusion. It is rather a summary of the reactions mentioned in the manuscript. Please, provide an appropriate conclusion section.

Additional comments

·         Lines 22 and 34. Write “originate” instead of “are originated”

·         A grammatic problem regards the use of “the” before nouns throughout the text. Delete the word “the” at least in the following lines: 33, 37, 79, 80, 88, 89, 192, 204, 215, 230, 242 and 263.

·         Line 38. There is a space after “-“ in β-glucosidase, which lacks an “o”.

·         Lines 48-49. Write “proteases” instead of “protease enzymes”.

·         Lines 58-60 must be deleted.

·         Line 63. Hydrogen peroxide is not produced by glucose oxidase, but after oxidation of glucose by this enzyme. Please, correct the sentence.

·         All figure captions need to have references unless they are yours exclusively.

·         Line 78 and 80. Delete “enzyme” after “catalase”.

·         Line 78. H2O2, numbers should be in subscripts

·         Line 879. “…microorganisms present in honey…”, not of honey.

·         Line 82. Delete “of the catalase enzyme”.

·         Lines 133-134. Invertase is not included in the European standard. It may be included in some standards, but not in the EU legislation.

·         Section 4.2. A lot of space is given to dicarbonyl compounds that are of importance mainly for Manuka honey, even though there are many other honeys where dicarbonyls are important. Please, enclose appropriate literature (e.g. doi:10.1021/jf903341t).

·         Lines 214-217. These two sentences should be one. Please, rephrase.

·         Line 227. “HMF” instead of “5-hydroxymethylfurfural”.

·         Lines 252-255. Even though HMF can have negative effects on humans, this happens in quite high concentrations, way higher that those found in honey. Please, report concentrations where these negative effects can occur.

·         Section 4.4. A scheme with the three stages of the Maillard reaction would be useful.

·         Line 276. Write “agents” instead of “ activities”.

Author Response

Dear Prof

Thank you very much for your valuable comments. I have strictly responded to all of your comments

Round 2

Reviewer 1 Report

Dear Authors,

Good revisions have been conducted and as we can see now that the revised manuscript presents in a good way.

However, in the abstract, the verb "originate" should be revised as "originated". The form or fashion of all Figures in the manuscript should be adjusted and well unified. Some Figures were vague and hardly to see clearly. Therefore, I hope the authors keep revising the paper.

Currently, I suggest "Minor revision" for this manuscript "molecules-1803874".

Best regards

Author Response

Dear reviewer

Really, I am very impressed by the level of your review and comments. I have learnt a lot from your review.

Reviewer 2 Report

The manuscript is significantly improved. The authors have responded adequately to my comments. The manuscript is to be published at its present form, after some minor corrections listed below.

·         Line 21. Delete “are” before “originate”

·         Line 58. Delete “…is so important since…”. Even though I agree that this article is important, authors themselves should not make this comment.

·         Line 59. Add “…that occur in honey” at the end of this sentence.

·         Delete the word “the” before nouns at lines: 132, 235, 299, 203

·         Table 1 should be moved right after it is mentioned in the text, that is after text Line 128.

·         Line 263. I would prefer “suggest” instead of “proved”.

·         Line 271. Write “Consequently” or “As a result”, not “However”.

·         Line 274-275. Delete “…, we discuss it here because there is some logic behind their assumes”.

·         Lines 288-290 are unnecessary.

·         Line 306. “melanoidin polymers”, not “melanoidins polymers”.

·         Line 310. The past form of the verb ‘lead” is ‘led”, not “leaded”

·         Table 1 contains a number of spelling mistakes. Please, consult an English speaker!

Author Response

Dear Prof

It is my pleasure to express my deep thanks for your valuable comments and corrections. I learnt a lot from your peer review. 

I owe you a great debt
